# 12*N*-Substituted Matrinol Derivatives Inhibited the Expression of Fibrogenic Genes via Repressing Integrin/FAK/PI3K/Akt Pathway in Hepatic Stellate Cells

**DOI:** 10.3390/molecules24203748

**Published:** 2019-10-17

**Authors:** Yunyang Bao, Yudong Pang, Sheng Tang, Tianyun Niu, Zhihao Guo, Hongwei He, Yinghong Li, Danqing Song

**Affiliations:** Institute of Medicinal Biotechnology, Chinese Academy of Medical Sciences and Peking Union Medical College, Beijing 100050, China; 17714333891@163.com (Y.B.); pyd101101@163.com (Y.P.); tang13874204108@163.com (S.T.); niutianyubasn@163.com (T.N.); guozhihao96@163.com (Z.G.); songdanqingsdq@hotmail.com (D.S.)

**Keywords:** liver fibrosis, matrinol, structure−activity relationship, COL1A1, integrin/FAK pathway

## Abstract

Twenty new 12*N*-substituted matrinol derivatives were synthesized and evaluated for their inhibitory effects on collagen α1 (I) (COL1A1) promotor in human hepatic stellate LX-2 cells. The structure-activity relationship (SAR) revealed that introducing a 12*N*-benzeneaminoacylmethyl substitution might significantly enhance the activity. Compound **8a** exhibited the highest inhibitory potency against COL1A1, and its inhibition activity against COL1A1 was further confirmed on both the mRNA and protein levels. It also effectively inhibited the expression of α smooth muscle actin (α-SMA), fibronectin and transforming growth factor β1 (TGFβ1), indicating an extensive inhibitory effect on the expression of fibrogenic genes. The primary mechanism study indicated that it might take action via the Integrin/FAK/PI3K/Akt signaling pathway. The results provided powerful information for further structure optimization, and compound **8a** was selected as a novel anti-fibrogenic lead for further investigation.

## 1. Introduction

Liver fibrosis refers to a reversible pathological process of an excessive precipitation of the hepatic diffuse extracellular matrix (ECM) in the liver which is caused by the abnormal proliferation of connective tissue. It seriously affects the function of liver, and is an inevitable stage for the development of chronic liver disease into fatal cirrhosis [1,2,3]. Progressive liver fibrosis can be triggered by a variety of factors, for example, chronic infection of various hepatitis viruses, alcoholism, nonalcoholic fatty liver disease (NAFLD)/nonalcoholic steatohepatitis (NASH) and other relatively rare diseases, such as autoimmune hepatitis, hemochromatosis, Wilson’s disease and congenital liver fibrosis [2]. Therefore, liver fibrosis affects a large population, and more seriously, the morbidity crowd displays a younger trend annually [4]. However, the treatment for liver fibrosis has been quite limited up to now [5]. Therefore, there is still a great demand to develop new anti-liver fibrosis candidates.

The activation, proliferation and transformation of hepatic stellate cells (HSCs) in the liver play vital roles in the process of fibrosis [6]. When liver injury occurs, HSCs release a series of fibrogenic factors, such as transforming growth factor β1 (TGFβ1) and fibronectin, to promote HSC activation [7,8,9]. In addition, they secrete an impressive amount of collagen type I (COL I, known as COL1A1 and COL1A2 in gene form), resulting in the excessive deposition of the extracellular matrix (ECM) [9]. Alpha smooth muscle actin (α-SMA, known as ATCA2 in gene form), the main cause of the pathological changes in multiple organs, are the main component of stellate cells-derived myofibroblasts [10]. Therefore, TGFβ1, fibronectin, COL1, and α-SMA are well accepted indicators of liver fibrogenesis. Among them, COL1 is the most important and indicative. It accounts for only 2–5% of the protein content in the normal liver, however, its content might increase to approximately 35% in the progress of liver fibrosis [11]. Therefore, a luciferase screening cell model based on COL1A1 promoter was established earlier by the authors [12], and was successfully applied to the screening and evaluation of anti-hepatic fibrosis drug candidates [13,14,15]. The compound stood out in an anti-COL1A1 assay and effectively reversed liver fibrosis in vivo [14].

The authors have been dedicated to the discovery of new anti-COL1A1 agents based on their innovative tricylic matrinic scaffold, a simplified structure of tetracyclic matrine (MT). MT, a quinolizidine alkaloid monomer isolated from *Sophora flavescens*, has been used as an anti-hepatitis B drug for decades in Chinese hospitals, and its anti-hepatic fibrosis effect has been gradually recognized and confirmed in clinics in recent years [16,17]. However, there is still room for the improvement in activity. In the authors’ previous study, by taking 12*N*-*p*-methyl benzenesulfonyl matrinic acid (**1**, Figure 1) as the lead, and taking advantage of bioisosterism between carboxylic acid and thiazole, the matrinic thiadiazole scaffold was constructed and identified to be a novel family of anti-fibrogenic agents [13], as depicted in Figure 1. The structure-activity relationship (SAR) demonstrated that suitable substitutions on the 12*N* atom or 11-terminal were beneficial for activity, indicating the promise of activity improvement through further structural modifications. Therefore, it might be interesting to conduct a new round of SAR toward modifiable sites of tricylic matrinic acid, aiming at developing them into a new class of anti-COL1A1 compounds.

In the present study, by taking **1** as the lead, the terminal carboxylic acid of 11-side chain was switched with methyl ester or hydroxyl, and meanwhile, the substituents with higher structural diversity were attached to 12*N* atom. Specifically, different linkers between the 12*N* atom and benzene ring as well as the varied substituents on the benzene ring were highlighted. Based on this strategy, as shown in Figure 1, a series of new methyl 12*N*-substituted matrinic butyrates and 12*N*-substituted matrinic butanols were designed, generated and evaluated for their ability to inhibit COL1A1 expression so as to replenish the anti-COL1A1 SAR and develop a novel class of anti-fibrotic agents. Herein, this study described the synthesis of 20 novel matrinic derivatives, the SAR analysis, anti- fibrogenic analysis and primary mechanism of action of the key compound.

## 2. Results and Discussion

### 2.1. Chemistry

A total of 24 tricyclic matrinic derivatives, of which 20 were firstly prepared from commercially available MT, purchased from Xi’an Tianbao Biotechnology Co., Ltd. (Shanxi, China) with purity of over 95%. The synthetic routes are depicted in Scheme 1.

Methyl 12*N*-matrinic butyrates (**3a–c**) were obtained by a 2-step procedure including hydrolysis and acylation in good yields. Further, 12*N*-matrinic butanols (**4a–c**) were obtained by a reduction reaction in the presence of lithium tetrahydroaluminum (LiAlH_4_) in an anhydrous condition from **3a–c** respectively in 80–90% yields [16]. The nucleophilic substitution reaction of bromoacetyl bromide **5** and substituted anilines in an equal molar ratio achieved key intermediates **6**, which underwent nucleophilic substitutions with **2** in a weak base to achieve methyl 12*N*-aminoacylmethyl substituted matrinic butyrates **7a–i** in yields of 30–50% [15]. Furthermore, 12*N*-aminoacylmethyl substituted matrinic butanols **8a–i** were gained from the reductions of 7**a–i** respectively in good yields, using LiAlH_4_ as the reductant. 

The desired products were purified with flash column chromatography on silica gel using dichloromethane/ methanol as an eluent.

### 2.2. Target Compounds Inhibited the Expresion of COL1A1 Promotor in Human Hepatic Stellate LX-2 Cells 

A single luciferase reporter gene detection model was applied to screen the inhibitory effects towards the COL1A1 promotor of all target compounds in the human hepatic stellate LX-2 cells at the concentration of 80 μM, taking EGCG as the positive control [18]. The LX-2 cells were transfected with COL1A1 promotor luciferase plasmid pGL4.17-COL1A1P by lipofactamine 2000 followed by the standard protocol (Invitrogen) for 24 h, then treated simultaneously with TGFβ1 and a tested compound for 24 h [12]. The structures and inhibitory effects (%) of all target compounds were shown in Table 1.

As a start, the 11-terminal carboxylic acid motif in lead **1** was replaced with methyl ester, and the effects of the different 12*N* substitutions were investigated, which were composed of two parts, the linker and benzene ring. Initially, various linkers, such as methyl, acyl, sulfonyl, aminoacylmethyl and methylaminoacylmethyl were applied while the bare benzene ring was retained. As depicted in Table 1, the introduction of benzyl (**3a**), benzoyl (**3b**) or benzenesulfonyl (**3c**) on the 12*N* atom only led to small fluctuations in the activity compared with lead **1**, while the introduction of aminoacylmethyl (**7a**) caused a significant increase in activity with the inhibition rate of 39.5%. However, the extension of the aminoacylmethyl linker by adding an extra methyl after the amino group caused a significant decrease in activity, as witnessed by the comparison of **7b** and **7a**, indicating the prevailing effect of benzeneaminoacylmethyl as the 12*N* substitution. 

Therefore, the aminoacylmethyl linker was then retained, and a series of methyl 12*N*-anilineacylmethyl matrinic butyrates **7c−i** were achieved with substituent variations on the benzene ring. As depicted in Table 1, the introduction of the halogen atom (**7c−e**), methyl (**7f**) or trifluoromethyl (**7g**) on the benzene ring showed less effect on the activity, while the introduction of nitro (**7h**) or 2,6-dimethyl (**7i**) gave negative effects. 

Next, 11-terminal carboxylic acid was changed into the methanol motif, and the corresponding matrinic butanols **4a−c** and **8a−i** were gathered respectively. Similarly, the introduction of 12*N*-benzyl (**4a**), benzoyl (**4b**) or benzenesulfonyl (**4c**) did not affect the activity much, and the introduction of 12*N*-aminoacylmethyl was beneficial for the activity. The corresponding compound **8a** gave the highest inhibitory rate of 51.0%, prevailing to **8b** with a methylaminoacylmethyl instead. Then, 12*N*-anilineacylmethyl matrinols **8c−i** were gained to investigate the impact of different substituents on the benzene ring. Similarly, the introduction of halogen atoms (**8c−e**) seemed to have a minimal effect on the activity. However, the introduction of methyl (**8f**), trifluoromethyl (**8g**), nitro (**8h**) and 2,6-dimethyl (**8i**) caused decreases in the activity to varying degrees. 

Compound **8a** displayed the highest anti-COL1A1 potency in the anti-COL1A1 screening, and was thus selected as the key compound to carry out further investigations.

### 2.3. **8a** Held a Good Safe Profile in LX-2 Cells

Before investigating the inhibitory effect of **8a** on liver fibrogenesis, its cytotoxicity in LX-2 cells at different concentrations and time durations were evaluated by sulforhodamine B (SRB) assay. **8a** hardly exerted any inhibition effect on the cell survival at the concentration of up to 200 μM (Figure 2A), and with the prolongation of the action time, **8a** did not show an obvious inhibitory effect on cell proliferation either (Figure 2B), indicationg that **8a** held a good safe profile in vitro.

### 2.4. **8a** Inhibited the Expressions of COL1A1 on mRNA and Protein Levels in Dose- and Time-dependent Manners

LX-2 cells were induced by TGFβ1 (2 ng/mL), and then treated with compounds **8a** at the concentration of 160 μM for 12 h, 24 h and 48 h, respectively. As anticipated, real-time PCR (RT-PCR) amplification results demonstrated that the stimulation of TGFβ1 caused a boomed expression of COL1A1, which was suppressed by the addition of **8a** after 24 and 48 h treatment, indicating a time-dependent inhibitory effect (Figure 3A). Next, the dose-dependent inhibitory behavior of **8a** on the COL1A1 mRNA level was investigated after a 24-h treatment, and it gave the inhibition rates of 56.3%, 95.0%, 100.1% and 105.9% at the concentrations of 40, 80, 160 and 200 μM, respectively (Figure 4A).

Further, its dose-dependent anti-COL1A1 effect on the protein level was carried out by a western blot assay. As indicated in Figure 5, **8a** significantly reversed the increase of the COL1A1 protein induced by TGFβ1 in a dose-dependent manner, with the inhibition rates of 40.3% and 48.3% at the concentrations of 160 and 200 μM, respectively. These results suggested that **8a** could effectively reduce the COL1A1 expression on both the mRNA and protein levels.

### 2.5. **8a** Inhibited the Expressions of α-SMA on mRNA and Protein Levels in Time- and Dose-dependent Manner

Then, the effects of **8a** on the ACTA2 mRNA expression and the α-SMA protein expression were evaluated by RT-PCR and western blot assays. As shown in Figure 3B, the enrichment of ACTA2 mRNA was significantly stimulated by the TGFβ1 treatment as expected, and the addition of **8a** substantially repressed this enrichment in a time-dependent and a dose-dependent manner as well, with the inhibition rate of 12.0%, 74.4%, 100.5% and 108.0% at the given concentrations, respectively (Figure 4B). However, **8a** also displayed a dose-dependent inhibition on the α-SMA expression, as disclosed by a western blot assay in Figure 5. Therefore, **8a** also inhibited the fibrogenic α-SMA expression. 

### 2.6. **8a** Inhibited the Expressions of Fibronectin on Protein Level in a Dose-Dependent Manner

Fibronectin is known to promote fibrosis in multiple organs. Therefore, it was considered as a fibrogenic factor worth considering [8,19]. As shown in Figure 5, upon the single irritation of TGFβ1, a 2.6-fold increase of the fibronectin expression was observed. The addition of **8a** dose-dependently reversed the elevation of the fibronectin level in a western blot analysis, and at the concentration of 200 μM. **8a** almost offset the increase in the fibronectin level caused by TGFβ1.

### 2.7. **8a** Inhibited the Expressions of TGFβ on mRNA and Protein Levels in Time and Dose-Dependent Manner

The previous mechanism studies revealed the positive regulation effects of TGFβ1 on a series of fibrosis genes like COL1A1, α-SMA as well as TGFβ1 itself. This is also known as a cascade amplification effect [20]. If the expression of TGFβ1 could be blocked by the target compound, then the cascade enlargement could be greatly suppressed. Therefore, the inhibitory effect of **8a** on TGFβ was investigated. As shown in Figure 3C, Figure 4C and Figure 5, TGFβ1 treatment successfully induced the expression booming of TGFβ1 itself on both the mRNA and protein levels, which were effectively reversed by the addition of **8a**. The RT-PCR analysis disclosed that **8a** reduced TGFβ1 mRNA levels in a time-dependent manner. **8a** also demonstrated the inhibitory effect on TGFβ1 mRNA in a dose-dependent manner, with the inhibitory rates of 19.4%, 36.6%, 64.6% and 71.8% at the given concentrations, respectively. A western blot analysis also disclosed a dose-dependent inhibition effect on TGFβ1 expression, with the inhibitory rate of 66.4% at the concentration of 200 μM. The blockage on the expression of TGFβ1 of **8a** could be greatly beneficial for its blockage on fibrogenesis.

Since **8a** exhibited comparable inhibitory effects at the concentrations of 160 μM and 200 μM, and significantly higher than the concentration of 80 μM, as shown by the results above, this study chose the concentration of 160 μM of **8a** for further examination.

### 2.8. **8a** Supressed the Intergrin-Focal Adhesion Pathway

To understand the antifibrogenic mechanism of **8a**, the transcriptome sequencing method was applied. The mRNAs of LX-2 cells were extracted, and the whole genome sequences were analyzed in three conditions (Figure 6A): (a) normal, (b) TGFβ1 treatment and (c) TGFβ1 and **8a** co-treatment. Then, the predicted and annotated genome sequences were analyzed for similarity with the Kyoto Encyclopedia of Genes and Genomes (KEGG) enzyme database followed by the assignment of each gene into the KEGG pathway charts. Based on individual analysis results of the KEGG pathways, the integrated biochemical pathway maps were constructed which demonstrated characteristic physiological features in the treatment of TGFβ1 or the co-treatment of TGFβ1 and **8a** (Figure 6B). The pathways with the highest gene changes were labeled as red, and 5 pathways were highlighted, including AGE-RAGE, TNF, protein digestion and absorption, focal adhesion (FA) and IL-17 (Figure 6B). Among them, the FA pathway is the mechanism most likely to affect liver fibrosis [21].

The integrins and focal adhesion kinase (FAK) constitute the key parts of the FA pathway [22]. The integrins are a major family of αβ heterodimeric, cell surface adhesion receptors for the extracellular matrix protein fibronectin and plays a fundamental role in transmitting signals between ECM ligand sites and their cytoplasmic domains [23]. The documented evidence supports that integrin αV, mainly expressed in liver, is closely associated with liver fibrogenesis [22]. Then, the inhibitory effects of **8a** on integrin αV expression were tested both on the protein and mRNA level. As depicted by a western blot assay in Figure 7A, the TGFβ1 treatment caused a significant increase in the integrin αV expression, while the addition of **8a** remarkably reversed this increase. However, **8a** hardly exerted any effect on the integrin αV expression without a TGFβ1 treatment. The changes in the mRNA expression of integrin αV were consistent with the protein expression change, as shown in Figure 7B. These results confirmed that **8a** could significantly inhibit the TGFβ1-reduced overexpression of integrin.

FAK is a non-receptor cytoplasmic protein, and activated integrin induces the activation of FAK by phosphorylation at Y397, and successively initiates the transduction of fibrogenic genes in HSCs [24]. The changes in the expression level of both active and inactive forms of FAK were monitored by a western blot assay in this study. The stimulation of TGFβ1 accelerated the phosphorylation of p-FAK (active form), and **8a** repressed the enrichment of p-FAK, while there was hardly any change on the expression of FAK, as depicted in Figure 7A. Therefore, this study concluded that **8a** probably inhibited the expression of liver fibrogenic genes via repressing the FA pathway.

FAK is a crucial FA protein that intersects many pathways and triggers the cellular response to ECM by acting as a signaling integrator. PI3K survival cascade can be activated by FAK Y397, leading to phosphorylation of AKT. In other words, FAK acts upstream of PI3K/Akt in a series of physiological change processes [22]. As shown in Figure 7A, **8a** distinctly repressed the phosphorylation of Akt in LX-2 cells after TGFβ1 stimulation. Therefore, it was speculated that **8a** might exert an anti-fibrotic effect through blocking of the integrin/FAK signaling pathway, then down-regulating the phosphorylation of PI3K/Akt in cascade, and finally inhibiting the expression of fibrogenic proteins, for example, COL1A1, α-SMA, fibronectin, as described in Figure 8.

## 3. Experimental Section

### 3.1. Apparatus, Materials, and Analysis Reagents

All chemical reagents and anhydrous solvents were obtained from commercial sources and used without further purification. The melting points (mp) were obtained with a MP90 melting point apparatus and were uncorrected (Mettler-Toledo, Greifensee, Switzerland). The ^1^H NMR and ^13^C NMR spectra were recorded on Bruker Avance (400 MHz, 500 MHz and 600MHz, Zürich, Switzerland) spectrometers (Bruker, Zürich, Switzerland), taking DMSO-*d_6_*, CDCl_3_, CD_3_COCD_3_ or CD_3_OD as the solvent, with Me_4_Si as the internal standard. The ESI high-resolution mass spectra (HRMS) were recorded on an AutospecUltima-TOF spectrometer (Micromass UK Ltd., Manchester, UK). Flash column chromatography was performed on Combiflash Rf 200 (Teledyne, NE, USA), particle size 0.038 mm.

The synthesis of target compounds **3a**, **3c**, **4a** and **4b** were reported previously [25].

### 3.2. Chemistry

#### 3.2.1. Synthesis of 12N-Benzoyl Matrinic Methyl Butyrate Hydrochloride (**3b**)

MT (5.0 g, 20 mmol) was added to 5 N NaOH solution (30 mL), and the reaction mixture was refluxed for 9 h, cooled to room temperature and then acidified with 2N HCl to pH 2−3. The solvent was removed in vacuo and the residue was dissolved with 2N HCl in methanol. Then, the reaction mixture was refluxed for 9 h. The solvent was removed by condensation to gain a yellow oil, which was suspended in ethanol (30 mL) and filtered to obtain intermediate **2** without further purification.

To a solution of **2** (3.0 g, 8.5 mmol) in dichloromethane, benzenesulfonyl chloride or benzoyl chloride or benzyl bromide (10.2 mmol, 1.2 eq) and triethylamine (1.3 g, 12.8 mmol, 1.5 eq) were added. The mixture was stirred at room temperature for 9 h until thin layer chromatography (TLC) indicated the completion of the reaction. Then, the mixture was washed with water (50 mL), saturated ammonium chloride solution (50 mL) and brine (50 mL), dried over anhydrous sodium sulfate, and then purified by flash column chromatography with dichloromethane/methanol as the eluent to give the desired product **3b** as a white solid, yield: 41%; m.p.: 88–90 °C. ^1^H NMR (600 MHz, CDCl_3_) *δ* 11.71–11.61 (br, 1H), 7.51 (d, *J* = 7.4 Hz, 2H), 7.43 (t, *J* = 7.4 Hz, 1H), 7.38 (t, *J* = 7.5 Hz, 2H), 4.34 (td, *J* = 11.0, 5.0 Hz, 1H), 4.20 (t, *J* = 13.4 Hz, 1H), 3.60 (s, 3H), 3.57–3.43 (m, 3H), 3.24–3.20 (m, 1H), 2.70–2.54 (m, 3H), 2.52–2.39 (m, 2H), 2.38–2.24 (m, 2H), 2.15 (t, *J* = 11.7 Hz, 2H), 1.96 (s, 1H), 1.90–1.83 (m, 1H), 1.76–1.56 (m, 6H), 1.46 (d, *J* = 14.3 Hz, 1H). ^13^C NMR (151 MHz, CDCl_3_) *δ* 174.5, 174.1, 136.1, 130.9, 128.8 (2), 128.2 (2), 65.9, 56.5, 56.4, 56.1, 51.5 (2), 37.7, 37.0, 34.0, 28.1, 25.8, 24.9, 21.8, 19.0, 18.7. ESI-HRMS: m/z Calcd for C_23_H_33_N_2_O_3_ [M + HCl − Cl]^+^, 385.2485; found, 385.2479.

#### 3.2.2. Synthesis of 12N-Benzenesulfonyl Matrinol Hydrochloride (**4c**)

To a solution of **3** (3.0 g, 8.5 mmol) in anhydrous THF at −10 °C, lithium aluminum hydride (10.2 mmol, 1.2 eq) was added, and stirred at –10 °C for 0.5 h until TLC showed the completion of the reaction. Then, the saturated ammonium chloride solution (2.0 mL) and methanol (5.0 mL) were added to quench the reaction. Then, dichloromethane (50 mL) was added. The mixture was stirred at room temperature for 15 min and filtered. The filtration was condensed, and the residue was purified by flash column chromatography with dichloromethane/methanol as the eluent to give the desired product **4c** as a pale yellow solid, yield: 90%; m.p.: 99–100 °C. ^1^H NMR (400 MHz, DMSO-*d_6_*) *δ* 10.73–10.13 (br, 1H), 7.85–7.76 (m, 2H), 7.70–7.62 (m, 1H), 7.62–7.53 (m, 2H), 4.10–3.99 (m, 1H), 3.83–3.63 (m, 2H), 3.63–3.49 (m, 1H), 3.49–3.24 (m, 1H), 3.19 (d, *J* = 11.4 Hz, 2H), 3.14–3.04 (m, 2H), 2.98–2.72 (m, 2H), 2.36–2.11 (m, 2H), 1.93–1.76 (m, 2H), 1.75–1.39 (m, 8H), 1.35–1.16 (m, 1H), 1.16–0.96 (m, 2H), 0.95–0.73 (m, 1H). ^13^C NMR (101 MHz, DMSO-*d_6_*) *δ* 141.9, 132.7, 129.3 (2), 126.5 (2), 63.0, 60.5, 57.9, 54.7, 54.5, 48.3, 38.1, 34.50, 32.2, 27.9, 24.9, 24.5, 22.3, 18.3, 18.1. ESI-HRMS: m/z Calcd for C_21_H_33_N_2_O_3_S [M + HCl – Cl]^+^, 393.2206; found, 393.2198.

#### 3.2.3. General Procedure for the Synthesis of Compounds **7a–i**

To a solution of bromoacetyl bromide (1.9 g, 9.0 mmol) in anhydrous dichloromethane, substituted aniline (9.9 mmol, 1.1 eq) and triethylamine (10.8 mmol, 1.2 eq) were added, and the mixture was stirred at room temperature for 2 h. Then, **2** (3.0 g, 9.0 mmol, 1.0 eq), potassium carbonate (3.7 g, 27 mmol, 3.0 eq) and acetonitrile (50 mL) were added, and the gained mixture was stirred at room temperature for 9 h until TLC showed the completion of the reaction. The solvent was evaporated, and the gained residue was dissolved in dichloromethane (50 mL) to form a yellow solution, which was washed with water (50 mL), saturated ammonium chloride solution (50 mL) and brine (50 mL) successively, then dried over anhydrous sodium sulfate and filtered. The filtration was condensed, and the residue was purified by flash column chromatography with dichloromethane/methanol as the eluent to give the desired products **7a–i**.

*Methyl 12N-(N-phenylacylmethylamine)matrinic butyrate* (**7a**) Total yield: 50%; white solid; m.p.: 139–141 °C. ^1^H NMR (400 MHz, DMSO-*d*_6_) *δ* 9.62 (s, 1H), 7.61 (d, *J* = 8.0 Hz, 2H), 7.31 (t, *J* = 7.8 Hz, 2H), 7.05 (t, *J* = 7.5 Hz, 1H), 3.52 (s, 3H), 3.43–3.34 (m, 1H), 3.15–2.88 (m, 3H), 2.81–2.60 (m, 2H), 2.41–2.32 (m, 1H), 2.32–2.20 (m, 2H), 1.98 (s, 1H), 1.96–1.88 (m, 1H), 1.88–1.70 (m, 3H), 1.70–1.47 (m, 6H), 1.47–1.35 (m, 3H), 1.35–1.17 (m, 3H). ^13^C NMR (101 MHz, DMSO-*d_6_*) *δ* 173.4, 169.6, 138.4, 128.7 (2), 123.4, 119.3 (2), 63.7, 56.9, 56.7, 55.5, 55.1, 53.4, 51.1, 37.3, 33.3, 32.8, 27.7, 27.5, 26.8, 21.1, 20.7, 19.1. ESI-HRMS: m/z Calcd for C_24_H_36_N_3_O_3_ [M + H]^+^, 414.2749; found, 414.2752.

*Methyl 12N-(N-benzylacylmethylamine)matrinic butyrate* (**7b**) Total yield: 30%; white solid; m.p.: 147–149 °C. ^1^H NMR (400 MHz,CD_3_OD) *δ* 7.35–7.20 (m, 5H), 4.51–4.34 (m, 2H), 3.62 (s, 3H), 3.48 (d, *J* = 16.6 Hz, 1H), 3.01–2.84 (m, 3H), 2.84–2.71 (m, 2H), 2.37 (dd, *J* = 12.0, 4.2 Hz, 1H), 2.32–2.23 (m, 2H), 2.13 (t, *J* = 3.1 Hz, 1H), 2.02–1.82 (m, 4H), 1.76–1.26 (m, 13H). ^13^C NMR (101 MHz, CD_3_OD) *δ* 175.7, 174.4, 140.1, 129.6 (2), 128.5 (2), 128.2, 65.8, 58.5, 58.3, 57.6, 55.6, 54.7, 52.0, 43.7, 39.3, 34.5, 34.4, 29.6, 28.9, 28.1, 22.4, 22.1, 20.9. ESI-HRMS: m/z Calcd for C_25_H_38_N_3_O_3_ [M + H]^+^, 428.2908; found, 429.2907.

*Methyl 12N-(N-p-fluorophenylacylmethylamine)matrinic butyrate* (**7c**) Total yield: 40%; white solid; m.p.: 138–139 °C. ^1^H NMR (600 MHz, CDCl_3_) *δ* 9.41 (s, 1H), 7.59–7.47 (m, 2H), 7.07–6.98 (m, 2H), 3.61 (s, 3H), 3.43 (d, *J* = 17.0 Hz, 1H), 3.16 (d, *J* = 11.7 Hz, 2H), 3.01 (d, *J* = 17.0 Hz, 1H), 2.81 (dd, *J* = 31.7, 11.1 Hz, 2H), 2.46 (dd, *J* = 12.3, 4.2 Hz, 1H), 2.33–2.26 (m, 2H), 2.09 (s, 1H), 2.00–1.82 (m, 4H), 1.74–1.36 (m, 12H). ^13^C NMR (151 MHz, CDCl_3_) *δ* 173.7, 169.8, 160.1, 158.5, 133.9, 133.9, 121.2, 121.2, 115.7, 115.5, 64.2, 57.4, 57.2, 56.5, 53.8, 51.5, 38.2, 33.9, 28.8, 27.89, 27.2, 21.4, 21.1, 20.4. ESI-HRMS: m/z Calcd for C_24_H_34_FN_3_O_3_ [M + H]^+^, 432.2657; found, 432.2648.

*Methyl 12N-(N-p-chlorophenylacylmethylamine)matrinic butyrate* (**7d**) Total yield: 44%; white solid; m.p.: 124–125 °C. ^1^H NMR (600 MHz, CDCl_3_) δ 9.46 (s, 1H), 7.54 (d, *J* = 8.3 Hz, 2H), 7.30–7.27 (m, 2H), 3.61 (s, 3H), 3.43 (d, *J* = 17.0 Hz, 1H), 3.15 (s, 1H), 3.01 (d, *J* = 17.0 Hz, 1H), 2.83 (d, *J* = 10.8 Hz, 1H), 2.78 (d, *J* = 11.3 Hz, 1H), 2.45 (dd, *J* = 12.8, 3.8 Hz, 1H), 2.29 (t, *J* = 7.2 Hz, 2H), 2.09 (s, 1H), 1.99–1.90 (m, 3H), 1.85 (d, *J* = 13.8 Hz, 1H), 1.73–1.35 (m, 13H). ^13^C NMR (151 MHz CDCl_3_) *δ* 173.7 (2), 169.9, 136.4, 129.0 (2), 120.7 (2), 64.1, 57.4, 57.2, 56.5, 53.8, 51.6 (2), 38.2, 33.9 (2), 28.7, 27.9, 27.2, 21.4, 21.1, 20.4. ESI-HRMS: m/z Calcd for C_24_H_34_ClN_3_O_3_ [M + H]^+^, 448.2361; found, 448.2357.

*Methyl 12N-(N-p-bromophenylacylmethylamine)matrinic butyrate* (**7e**) Total yield: 45%; white solid; m.p.:151–152 °C. ^1^H NMR (600 MHz, CDCl_3_) δ 9.46 (s, 1H), 7.54–7.38 (m, 4H), 3.61 (s, 3H), 3.43 (d, *J* = 17.0 Hz, 1H), 3.19–3.11 (m, 2H), 3.00 (d, *J* = 17.0 Hz, 1H), 2.80 (m, 2H), 2.45 (dd, *J* = 12.2, 4.2 Hz, 1H), 2.29 (t, *J* = 7.2 Hz, 2H), 2.09 (s, 1H), 2.00–1.82 (m, 4H), 1.73–1.37 (m, 12H). ^13^C NMR (151 MHz, CDCl_3_) *δ* 173.8 (2), 170.1, 137.0, 132.0 (2), 121.2 (2), 116.7, 64.2, 57.5, 57.3, 56.6, 53.9, 51.7 (2), 38.3, 34.0, 28.8, 28.0, 27.3, 21.5, 21.2, 20.5. ESI-HRMS: m/z Calcd for C_24_H_34_BrN_3_O_3_ [M + H]^+^, 492.1856; found, 492.1851.

*Methyl 12N-(N-p-methylphenylacylmethylamine)matrinic butyrate* (**7f**) Total yield: 44%; pale yellow solid; m.p.: 128–131 °C. ^1^H NMR (500 MHz, DMSO-*d*_6_) *δ* 9.55 (s, 1H), 7.49 (d, *J* = 8.4 Hz, 2H), 7.10 (d, *J* = 8.2 Hz, 2H), 3.52 (s, 3H), 3.39–3.31 (m, 1H), 3.00 (m, 3H), 2.70 (m, 2H), 2.39–2.27 (m, 3H), 2.25 (s, 3H), 1.99–1.74 (m, 5H), 1.61–1.23 (m, 12H). ^13^C NMR (126 MHz, DMSO-*d_6_*) *δ* 173.5, 169.4, 136.0, 132.3, 129.1 (2), 119.4 (2), 63.7, 56.9, 56.7, 55.5, 55.1, 53.4, 51.2, 37.3, 33.2, 32.9, 27.7, 27.5, 26.9, 21.1, 20.7, 20.5, 19.1. ESI-HRMS: m/z Calcd for C_25_H_38_N_3_O_3_ [M + H]^+^, 428.2907; found, 428.2898.

*Methyl 12N-(N-p-trifluoromethylphenylacylmethylamine)matrinic butyrate* (**7g**) Total yield: 34%; white solid; m.p.: 169–171 °C. ^1^H NMR (500 MHz, DMSO-*d*_6_) δ 9.99 (s, 1H), 7.86 (d, *J* = 8.5 Hz, 2H), 7.67 (d, *J* = 8.6 Hz, 2H), 3.50 (s, 3H), 3.40 (d, *J* = 16.5 Hz, 1H), 3.10–3.00 (m, 3H), 2.68 (dt, *J* = 27.3, 13.6 Hz, 2H), 2.39–2.32 (m, 1H), 2.26 (t, *J* = 6.8 Hz, 2H), 1.97 (t, *J* = 3.0 Hz, 1H), 1.91 (d, *J* = 11.8 Hz, 1H), 1.80 (m, 3H), 1.58–1.23 (m, 12H). ^13^C NMR (126 MHz, DMSO-*d_6_*) *δ* 173.5, 170.6, 142.1, 126.0, 126.0, 126.0, 119.3 (2), 63.7, 56.9, 56.7, 55.6 (2), 55.5, 53.5, 51.2, 37.3, 33.3, 33.0, 27.7, 27.5, 26.8, 21.1, 20.7, 19.0. ESI-HRMS: m/z Calcd for C_25_H_35_F_3_N_3_O_3_ [M + H]^+^, 482.2625; found, 482.2613.

*Methyl 12N-(N-p-nitrophenylacylmethylamine)matrinic butyrate* (**7h**) Total yield: 40%; pale yellow solid; m.p.:183–185 °C. ^1^H NMR (600 MHz,CDCl_3_) δ 9.86 (s, 1H), 8.21 (d, *J* = 8.6 Hz, 2H), 7.77 (d, *J* = 8.7 Hz, 2H), 3.60 (s, 3H), 3.46 (d, *J* = 17.2 Hz, 1H), 3.18 (d, *J* = 11.7 Hz, 2H), 3.05 (d, *J* = 17.1 Hz, 1H), 2.80 (m, 2H), 2.43 (dd, *J* = 12.4, 4.1 Hz, 1H), 2.29 (t, *J* = 7.2 Hz, 2H), 2.09 (d, *J* = 3.5 Hz, 1H), 1.93 (m, 3H), 1.83 (d, *J* = 13.8 Hz, 1H), 1.72–1.51 (m, 8H), 1.42 (m, 4H). ^13^C NMR (151 MHz, CDCl_3_) *δ* 173.9, 170.8, 143.7, 143.6, 125.2 (2), 119.1 (2), 72.1, 64.2, 60.6, 57.5, 57.3, 56.5, 53.9, 51.7, 38.3, 33.9, 28.8, 28.0, 27.4, 21.5, 21.2, 20.5. ESI-HRMS: m/z Calcd for C_24_H_35_N_4_O_5_ [M + H]^+^, 459.2602; found, 459.2593.

*Methyl 12N-(N-2,6-dimethylphenylacylmethylamine)matrinic butyrate hydrochloride* (**7i**) Total yield: 33%; pale yellow solid; m.p.: 228–230 °C. ^1^H NMR (500 MHz, DMSO-*d*_6_) δ 11.12 (br, *J* = 8.9 Hz, 1H), 10.41 (br, *J* = 8.9 Hz, 1H), 10.37 (s, 1H), 7.10 (q, *J* = 4.5 Hz, 3H), 4.69 (d, *J* = 15.6 Hz, 1H), 4.30 (t, *J* = 10.3 Hz, 1H), 4.14 (m, 2H), 3.66 (d, *J* = 10.4 Hz, 1H), 3.60 (s, 3H), 3.52–3.45 (m, 2H), 3.26 (t, *J* = 11.0 Hz, 2H), 3.00 (m, 2H), 2.66–2.58 (m, 1H), 2.45 (t, *J* = 7.1 Hz, 3H), 2.21 (s, 6H), 1.95 (d, *J* = 12.8 Hz, 1H), 1.84–1.61 (m, 10H).^13^C NMR (126 MHz, DMSO-*d_6_*) *δ* 176.26, 162.8, 135.0 (2), 133.8, 127.9 (2), 127.0, 60.6, 60.2, 59.9, 55.9, 54.2, 54.1, 52.1, 35.9, 31.8, 30.2, 28.3, 24.0, 23.6, 22.6, 18.6, 18.3, 18.0, 18.0. ESI-HRMS: m/z Calcd for C_26_H_40_N_3_O_3_ [[M + HCl – Cl]^+^, 442.3064; found, 442.3056.

#### 3.2.4. General Procedure for the Synthesis of Compounds **8a–i**

Following the procedure in 3.2.2, to a solution of **7** (8.5 mmol) in anhydrous THF at −10 °C, lithium aluminum hydride (10.2 mmol, 1.2 eq) was added, and stirred at –10 °C for 0.5 h until TLC showed the completion of the reaction. The saturated ammonium chloride solution (2.0 mL) and methanol (5.0 mL) were added to quench the reaction, and then dichloromethane (50 mL) was added. The mixture was stirred at room temperature for 15 min and filtered. The filtration was condensed, and the residue was purified by flash column chromatography with dichloromethane/methanol as the eluent to give the desired products **8a–i**, respectively.

*12N-(N-Phenylacylmethylamine)matrinol* (**8a**) Total yield: 80%; pale yellow solid; m.p.: 138–140 °C. ^1^H NMR (400 MHz, DMSO-*d*_6_) *δ* 9.63 (s, 1H), 7.68–7.52 (m, 2H), 7.38–7.22 (m, 2H), 7.15–6.90 (m, 1H), 4.32 (t, *J* = 5.0 Hz, 1H), 3.34 (t, *J* = 5.2 Hz, 3H), 3.10–2.96 (m, 3H), 2.79–2.62 (m, 2H), 2.38 (dd, *J* = 11.6, 4.1 Hz, 1H), 2.01–1.96 (m, 1H), 1.96–1.88 (m, 1H), 1.88–1.73 (m, 3H), 1.65–1.45 (m, 4H), 1.45–1.20 (m, 10H). ^13^C NMR (101 MHz, DMSO-*d*_6_) *δ* 169.7, 138.4, 128.7 (2), 123.4, 119.3 (2), 63.8, 60.7, 56.9, 56.7, 55.9, 55.2, 53.4, 37.4, 32.8, 32.7, 28.2, 27.7, 27.0, 21.1, 20.8, 20.3.ESI-HRMS: m/z Calcd for C_23_H_36_N_3_O_2_ [M + H]^+^, 386.2802; found, 386.2789.

*12N-(N-Benzylacylmethylamine)matrinol* (**8b**) Total yield: 74%; white solid; m.p.: 124–126 °C. ^1^H NMR (400 MHz, CD_3_OD) *δ* 7.35–7.23 (m, 5H), 4.52–4.32 (m, 2H), 3.51–3.38 (m, 3H), 3.35–3.31 (m, 1H), 3.00–2.71 (m, 5H), 2.38 (dd, *J* = 12.0, 4.3 Hz, 1H), 2.13 (s, 1H), 2.03–1.83 (m, 4H), 1.76–1.29 (m, 15H). ^13^C NMR (101 MHz, CD_3_OD) *δ* 174.4, 140.1, 129.6 (2), 128.5 (2), 128.3 (2), 65.8, 62.8, 58.5, 58.3, 57.9, 55.7, 54.6, 43.7, 39.5, 34.3, 33.8, 30.4, 28.9, 28.2, 22.4, 22.1. ESI-HRMS: m/z Calcd for C_24_H_38_N_3_O_2_ [M + H]^+^, 400.2958; found, 400.2951.

*12N-(N-p-Fluorophenylacylmethylamine)matrinol* (**8c**) Total yield: 90%; white solid; m.p.: 150–151 °C. ^1^H NMR (600 MHz, DMSO-*d_6_*) *δ* 9.68 (s, 1H), 7.64 (s, 2H), 7.14 (t, *J* = 8.6 Hz, 2H), 4.35–4.28 (m, 1H), 3.37–3.32 (m, 2H), 3.10–2.95 (m, 3H), 2.70 (dd, *J* = 31.4, 11.0 Hz, 2H), 2.36 (dd, *J* = 11.6, 4.0 Hz, 1H), 2.00–1.88 (m, 2H), 1.87–1.73 (m, 3H), 1.57–1.26 (m, 15H).^13^C NMR (151 MHz, DMSO-*d_6_*) *δ* 169.7, 158.9, 157.3, 134.9, 121.2, 121.1, 115.3, 115.1, 63.8, 60.7, 56.9, 56.7, 56.0, 55.3, 53.5, 37.4, 32.7, 28.3, 27.7, 26.9, 21.1, 20.8, 20.3. ESI-HRMS: m/z Calcd for C_23_H_34_FN_3_O_2_ [M + H]^+^, 404.2708; found, 404.2703.

*12N-(N-p-Chlorophenylacylmethylamine)matrinol* (**8d**) Total yield: 89%; pale solid; m.p.: 150–151 °C. ^1^H NMR (600 MHz,CDCl_3_) *δ* 9.51 (s, 1H), 7.53 (d, *J* = 8.4 Hz, 2H), 7.30–7.27 (m, 2H), 3.62–3.53 (m, 2H), 3.41 (d, *J* = 17.1 Hz, 1H), 3.16 (s, 2H), 3.02 (d, *J* = 17.1 Hz, 1H), 2.87–2.74 (m, 2H), 2.45 (dd, *J* = 12.3, 4.2 Hz, 1H), 2.09 (s, 1H), 2.01–1.88 (m, 3H), 1.84 (d, *J* = 13.7 Hz, 1H), 1.70–1.35 (m, 15H). ^13^C NMR (151 MHz, CDCl_3_) *δ* 170.3, 136.5, 129.1 (2), 120.8 (2), 64.3, 62.7, 60.5, 57.5, 57.3, 56.9, 54.0, 38.5, 32.8, 29.8, 29.3, 28.0, 27.4, 21.5, 21.3, 21.2, 14.3. ESI-HRMS: m/z Calcd for C_23_H_34_ClN_3_O_2_ [M + H]^+^, 420.2412; found,420.2405.

*12N-(N-p-Bromophenylacylmethylamine)matrinol* (**8e**) Total yield: 85%; pale yellow solid; m.p.: 159–161 °C. ^1^H NMR (600 MHz, CD_3_COCD_3_) *δ* 9.75 (s, 1H), 7.63–7.59 (m, 2H), 7.50–7.46 (m, 2H), 4.29 (t, *J* = 5.1 Hz, 1H), 3.35–3.31 (m, 3H), 3.09–2.98 (m, 3H), 2.70 (m, 2H), 2.36 (dd, *J* = 11.7, 4.1 Hz, 1H), 1.98 (d, *J* = 9.4 Hz, 1H), 1.95–1.88 (m, 1H), 1.85–1.75 (m, 3H), 1.63–1.22 (m, 14H).^13^C NMR (151 MHz, CD_3_COCD_3_) *δ* 170.0, 137.8, 131.4, 121.3, 114.9, 63.7, 60.6, 59.7, 56.8, 56.6, 56.0, 55.4, 53.4, 37.4, 32.8, 32.7, 28.2, 27.6, 26.9, 21.1, 20.8, 20.3, 14.1. ESI-HRMS: m/z Calcd for C_23_H_34_BrN_3_O_2_ [M + H]^+^, 464.1907; found, 464.1907.

*12N-(N-p-Methylphenylacylmethylamine)matrinol hydrochloride* (**8f**) Total yield: 76%; white solid; m.p.: 186–188 °C. ^1^H NMR (500 MHz, DMSO-*d*_6_) *δ* 11.13 (br, 1H), 10.88 (br, 1H), 10.18 (br, 1H), 7.53 (d, *J* = 8.2 Hz, 2H), 7.16 (d, *J* = 8.2 Hz, 2H), 4.63–4.55 (m, 1H), 4.24–4.14 (m, 1H), 4.14–3.99 (m, 2H), 3.66–3.59 (m, 1H), 3.51–3.36 (m, 4H), 3.25 (t, *J* = 11.3 Hz, 2H), 3.06–2.87 (m, 2H), 2.62–2.53 (m, 1H), 2.42 (d, *J* = 12.4 Hz, 1H), 2.26 (s, 3H), 1.89–1.59 (m, 10H), 1.49–1.33 (m, 4H). ^13^C NMR (126 MHz, DMSO-*d*_6_) *δ* 162.6, 135.6, 133.3, 129.4 (2), 119.5 (2), 60.4, 60.2, 59.9, 56.4, 54.2, 54.2, 51.8, 36.0, 31.8, 30.2, 28.7, 23.9, 23.6, 23.1, 20.6, 18.0, 17.9. ESI-HRMS: m/z Calcd for C_24_H_38_N_2_O_3_ [M + HCl – Cl]^+^, 400.2959; found, 400.2958;

*12N-(N-p-Trifluoromethylphenylacylmethylamine)matrinol* (**8g**) Total yield: 78%; white solid; m.p.: 181–183 °C. ^1^H NMR (500 MHz, DMSO-*d*_6_) *δ* 10.00 (br, 1H), 7.86 (d, *J* = 8.5 Hz, 2H), 7.68 (d, *J* = 8.5 Hz, 2H), 4.33 (t, *J* = 5.0 Hz, 1H), 3.39 (s, 1H), 3.31 (q, *J* = 5.9 Hz, 2H), 3.15–2.98 (m, 3H), 2.71 (m, 2H), 2.38 (dd, *J* = 11.3, 4.1 Hz, 1H), 1.97 (s, 1H), 1.94–1.88 (m, 1H), 1.88–1.72 (m, 3H), 1.60–1.25 (m, 14H).^13^C NMR (126 MHz, DMSO-*d*_6_) *δ* 170.6, 142.1, 126.1, 126.1, 126.0, 126.0, 125.5, 119.3, 63.8, 60.7, 56.9, 56.7, 55.9, 55.5, 53.5, 37.4, 33.0, 32.7, 28.2, 27.7, 27.0, 21.1, 20.8, 20.2. ESI-HRMS: m/z Calcd for C_24_H_35_F_3_N_3_O_2_ [M + H]^+^, 454.2675; found, 454.2663.

*12N-(N-p-Nitrophenylacylmethylamine)matrinol* (**8h**) Total yield: 80%; white solid; m.p.: 169–171 °C. ^1^H NMR (500 MHz, DMSO-*d*_6_) *δ* 10.23 (br, 1H), 8.28–8.17 (m, 2H), 7.96–7.87 (m, 2H), 4.32 (t, *J* = 5.0 Hz, 1H), 3.39 (d, *J* = 16.6 Hz, 1H), 3.29 (q, *J* = 6.0 Hz, 2H), 3.16–3.01 (m, 3H), 2.76–2.65 (m, 2H), 2.38 (dd, *J* = 11.5, 4.1 Hz, 1H), 1.96 (t, *J* = 3.0 Hz, 1H), 1.91 (m, 1H), 1.87–1.74 (m, 3H), 1.61–1.23 (m, 14H). ^13^C NMR (126 MHz, DMSO-*d*_6_) *δ* 171.0, 144.7, 142.3, 125.0 (2), 119.1(2), 63.8, 60.7, 56.9, 56.7, 56.0, 55.8, 53.6, 37.4, 33.1, 32.7, 28.2, 27.7, 27.0, 21.1, 20.8, 20.1. ESI-HRMS: m/z Calcd for C_23_H_35_N_4_O_4_ [M + H]^+^, 431.2653; found, 431.2654.

*12N-(N-2,6-Dimethylphenylacylmethylamine)matrinol hydrochloride* (**8i**) Total yield: 75%; white solid; m.p.: 199–200 °C. ^1^H NMR (500 MHz, DMSO-*d*_6_) *δ* 11.14 (br, 1H), 10.36 (s, 1H), 10.27 (br, 1H), 7.10 (q, *J* = 4.5 Hz, 3H), 4.51 (s, 3H), 4.30–4.08 (m, 3H), 3.65 (m, 1H), 3.51–3.41 (m, 2H), 3.26 (t, *J* = 11.0 Hz, 2H), 3.06–2.92 (m, 2H), 2.65–2.58 (m, 1H), 2.43 (d, *J* = 12.3 Hz, 1H), 2.19 (d, *J* = 9.7 Hz, 6H), 1.90–1.57 (m, 10H), 1.45 (m, 4H). ^13^C NMR (126 MHz, DMSO-*d*_6_) *δ* 162.8, 135.0 (2), 133.8, 127.9 (2), 127.0, 60.6, 60.2, 59.9, 55.9, 54.2, 54.1, 52.1, 35.9, 31.8, 30.2, 28.3, 24.0, 23.6, 22.6, 18.6, 18.3, 18.1, 18.0. ESI-HRMS: m/z Calcd for C_25_H_40_N_3_O_2_ [M + HCl – Cl]^+^, 414.3115; found, 414.3109.

### 3.3. Biology Assay

#### 3.3.1. Cell Culture and Screening of Compounds

The cells were laid on a 96-well plate, cultured in Dulbecco’s Modified Eagle’s medium (DMEM), containing 10% fetal bovine serum (FBS) in a 5% CO_2_ atmosphere at 37 °C. Moreover, the serum-free culture was required until the cells reached 90%−95% confluence. After 24 h, the cells were treated with a matrinic derivative (80 μM) for 24 h. The COL1A1 promotor activity was determined using the Bright-Glo luciferase assay system.

#### 3.3.2. Sulforhodamine B (SRB) Assay

The LX-2 cells were seeded in a 96-well plate, cultured in DMEM, containing 10% fetal bovine serum (FBS) in a 5% CO_2_ atmosphere at 37 °C. After the treatment of **8a** at the given concentrations for a given time duration, and washing three times with PBS, the cell monolayers were fixed with 10% (wt/vol) trichloroacetic acid for 1 h, and stained with SRB for 30 min, after which the excess dye was removed by washing repeatedly with 1% (vol/vol) acetic acid. The protein-bound dye was dissolved in 10 mM Tris base solution for OD determination at 510 nm using a microplate reader.

#### 3.3.3. RT-qPCR Assay

The LX-2 cells were seeded in a 6-well plate, cultured in DMEM, containing 10% fetal bovine serum (FBS) in a 5% CO_2_ atmosphere at 37 °C. A serum-free culture was required until the cells reached 90%−95% confluence. After 24 h, the cells were treated with TGFβ1 (2 ng/mL) and matrinic derivatives (80 μM) for 24 h. The total RNA from the LX-2 cells was extracted using Trizol reagent, purified by NucleoSpin RNA Clean-up. The reverse transcription was performed with a Transcriptor first strand cDNA synthesis kit. The cDNA was then analyzed by ABI 7500 Fast Real-Time PCR System using TaqMan probes of TGFβ1, COL1A1, α-SMA, and GAPDH (sequence reserved by ABI) and FastStart Universal Probe master mix (Roche).

#### 3.3.4. Western Blot

The LX-2 cells were cultured as described above. Briefly, the cells were washed with phosphate-buffered saline (PBS) and were lysed in radioimmunoprecipitation assay (RIPA) lysis for 30 min in 4 °C. The supernatant was collected after centrifugation at 12000 g, 4 °C for 15 min. Equal amounts of the protein were quantified with a Bradford assay, separated by SDS-PAGE and transferred to polyvinylidene difluoride membranes. The membranes were blocked for one hour at room temperature in PBST containing 5% milk and probed with specific first antibodies overnight at room temperature. The membrane was washed 3 times by PBST, followed by horseradish peroxidase-conjugated secondary antibodies and GAPDH. The proteins were visualized using chemiluminescence reagents. The antibodies used in the western blot analysis were obtained from Abcam(anti-collagen 1(ab34710), anti-alpha smooth muscle actin (ab32575), anti-TGF beta 1 antibody(ab179695), anti-fibronectin(ab32419)), Cell Signaling Technology(GAPDH(D16H11) Rabbit mAb(5174), Phospho-FAK(Tyr397)Antibody(3283S), FAK Antibody(3285S), Phospho-Akt (Ser473)(D9E) Rabbit mAb(4060S)), BD biosciences (Purified mouse anti-CD51, (611012)), Proteintech (AKT Antibody Rabbit Polyclonal, (10176-2-AP)).

#### 3.3.5. Total mRNA Extraction

The LX-2 cells were seeded in a cell culture dish (100 mm × 20 mm style), cultured in DMEM, containing 10% fetal bovine serum (FBS) in a 5% CO_2_ atmosphere at 37 °C. A serum-free culture was required until the cells reached 90%−95% confluence. After 24 h, the cells were treated with DMSO, TGFβ1 (2 ng/mL) or TGFβ1 (2ng/mL) and **8a** (160 μM) for 24 h. The total RNA from the LX-2 cells was extracted using Trizol reagent, purified by a NucleoSpin RNA clean-up kit (Macherey-Nagel, Germany).

#### 3.3.6. Genome Sequencing and Annotation

All the mRNA samples were used for a global transcriptome analysis by Annoroad Co. The significantly differentially expressed genes were identified when compared with the normalized read counts between the vehicle, TGFβ1 treatment, **8a** and TGFβ1 co-treatment groups with *p* < 0.05 and |Log2FoldChange| > 1.

#### 3.3.7. Gene Network/Pathway Analysis

KEGG (Kyoto Encyclopedia of Genes and Genomes, Kyoto Encyclopedia of Genes and Genomes) is a database for genome deciphering. The predicted and annotated gene sequences were analyzed for similarity with the KEGG enzyme database followed by the assignment of each gene into the KEGG (Kyoto Encyclopedia of Genes and Genomes) pathway chart. Based on individual analysis results of the KEGG pathways, the integrated biochemical pathway maps were constructed which demonstrated the characteristic physiological features in the hepatic fibrosis. The existence of a certain pathway was then determined and integrated when the component genes within the corresponding pathway had been completely identified.

#### 3.3.8. Statistics

The results are presented as the mean values ± standard error of the independent triplicate experiments. All statistical analyses were performed by using a two-tailed Student’s t-test and the *p*-values of less than 0.05 were considered statistically significant.

## 4. Conclusions

To summarize, a series of novel matrinic derivatives were designed, synthesized and evaluated for their inhibitory effect on the COL1A1 promotor. The SAR indicated that the introduction of benzeneaminoacylmethyl motif on the 12*N* atom was favorable for the activity. Among them, compound **8a** gave the highest inhibitory effect on the COL1A1 promotor at the rate of 51.0% on the cellular level at the concentration of 80 μM. Its inhibition activity against COL1A1 was further confirmed on both the mRNA and protein levels. Additionally, it effectively inhibited the expression of a series of fibrogenic proteins, such as α-SMA, fibronectin and TGFβ, indicating a promise against liver fibrogenesis. Further study indicated that it might exert the anti-fibrogenic effect via repressing the Integrin/FAK/PI3K/Akt pathway. Overall, this study greatly enriched the anti-COL1A1 SAR of the tricyclic matrinic derivatives and thus offered powerful information for further structure optimization, noting that compound **8a** was chosen as an ideal anti-liver fibrosis lead. This study provided a deep understanding on the anti-fibrogenic mechanism of these matrinic analogues.

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
