# Peer review of "12*N*-Substituted Matrinol Derivatives Inhibited the Expression of Fibrogenic Genes via Repressing Integrin/FAK/PI3K/Akt Pathway in Hepatic Stellate Cells"

_molecules, 2019, doi:10.3390/molecules24203748_

Round 1

Reviewer 1 Report

The authors have improved the content of manuscript "12N-substituted matrinol derivatives inhibited the expression of fibrogenic genes via repressing integrin/FAK/PI3K/Akt pathway in hepatic stellate cells" by performing additional experiments.

The authors have used the "Track changes" option which I thought was to highlight the introduction of new figures and content. But comment 16 seems to be written in Chinese language which I do not understand. Comment 17 is also confusing. Many oddly-worded sentences and grammatical concerns remain. For example, "It was delightful to see that there was no cytotoxicity......" The authors should do a thorough proof-reading of the manuscript.

Author Response

Thank you very much for the constructive advice. We are very sorry for the mistakes and confusions appeared in our earlier version, and we made revisions accordingly. We have changed the sentence "It was delightful to see that there was no cytotoxicity......" into “8a hardly exerted any inhibition effect on cell survival at the concentration up to 200 μM”, and et al. And no confusion on the tracking will be found this time.

Reviewer 2 Report

This study shows the effected of a selected new 12N-substituted matrinol derivatives, named 8a and inhibiting COL1A1 promotor transcriptional activity in human hepatic stellate LX-2 cells. Compound 8a also effectively inhibited the expression of α-SMA, fibronectin and TGFβ1, indicating powerful anti-fibrogenic effects, mediated through integrin/FAK/PI3K/Akt signaling pathway. These results encourage further investigation of 8a as a novel anti-fibrogenic agent. 

The manuscript is well structured, data are convincing and support the conclusions. Overall it is a very good work. However, LX-2 are notoriously the worst model to study HSC and fibrosis in vitro. Whereas the model is appropriate for screening purposes, as in this work, key findings with selected doses and times of administration must be replicated in primary human hepatic stellate cells, which are also available commercially.

Author Response

Thanks for your very constructive advice. We are now on the way to construct the human primary hepatic stellate cell model, and the experiments are actively on ongoing, and the results will be presented in our future papers.

Round 2

Reviewer 2 Report

None